# MQTransformer: Multi-Horizon Forecasts with Context Dependent and Feedback-Aware Attention

## Abstract

Recent advances in neural forecasting have produced major improvements in accuracy for probabilistic prediction. In this work, we propose novel improvements to the current state of the art by incorporating changes inspired by recent advances in Transformer architectures for Natural Language Processing. We develop a novel decoder-encoder attention for context-alignment, improving forecasting accuracy by allowing the network to study its own history based on the context for which it is producing a forecast. We also present a novel positional encoding that allows the neural network to learn context-dependent seasonality functions as well as arbitrary holiday distances. Finally, we show that the current state of the art MQ-Forecaster (Wen et al., 2017) models display excess variability by failing to leverage previous errors in the forecast to improve accuracy. We propose a novel decoder-self attention scheme for forecasting that produces significant improvements in the excess variation of the forecast.

## 1 Introduction

Time series forecasting is a fundamental problem in machine learning with relevance to many application domains including supply chain management, finance, healthcare analytics, and more. Modern forecasting applications require predictions of many correlated time series over multiple horizons. In multi-horizon forecasting, the learning objective is to produce forecasts for multiple future horizons at each time-step. Beyond simple point estimation, decision making problems require a measure of uncertainty about the forecasted quantity. Access to the full distribution is usually unnecessary, and several quantiles are sufficient (many problems in Operations Research use the $50^{th}$ and $90^{th}$ percentiles, for example).

As a motivating example, consider a large e-commerce retailer with a system to produce forecasts of the demand distribution for a set of products at a target time $T$. Using these forecasts as an input, the retailer can then optimize buying and placement decisions to maximize revenue and/or customer value. Accurate forecasts are important, but – perhaps less obviously – forecasts that don't exhibit excess volatility as a target date approaches minimize costly, bull-whip effects in a supply chain (Chen et al., 2000; Bray and Mendelson, 2012).

Recent work applying deep learning to time-series forecasting focuses primarily on the use of recurrent and convolutional architectures (Nascimento et al., 2019; Yu et al., 2017; Gasparin et al., 2019; Mukhoty et al., 2019; Wen et al., 2017)[1]. These are Seq2Seq architectures (Sutskever et al., 2014) – which consist of an *encoder* that summarizes an input sequence into a fixed-length context vector, and a *decoder* which produces an output sequence. This line of work has lead to major advances in forecast accuracy for complex problems, and real-world forecasting systems increasingly rely on neural nets. Accordingly, a need for black-box forecasting system diagnostics has arisen. Foster and Stine (2021) use probabilistic martingales to study the dynamics of forecasts produced by an arbitrary forecasting system. They can be used to detect the degree to which forecasts adhere to the martingale model of forecast evolution (Heath and Jackson, 1994) and to detect unnecessary volatility (above and beyond any inherent uncertainty) in the forecasts produced – that is to say, Foster

---

[1]For a complete overview see Benidis et al. (2020)

and Stine (2021) describe a way to connect the excess variation of a forecast to accuracy misses against the realized target. While tools such as martingale diagnostics can be used to detect flaws in forecasts, the question of how to incorporate that information into model design is unexplored. Existing multi-horizon forecasting architectures (Wen et al., 2017; Madeka et al., 2018) are designed to minimize quantile loss, but they do not explicitly handle excess variation, since forecasts on any particular date are not made aware of errors in the forecast for previous dates.

Another limitation in many existing architectures is the well known information bottleneck, where then encoder transmits information to the decoder via a single hidden state. To address this, Bahdanau et al. (2014) introduce a method called *attention*, allowing the decoder to take as input a weighted combination of relevant latent encoder states at each output time step, rather than using a single context to produce all decoder outputs. While NLP is the predominate application of attention architectures to date, in this paper we use novel attention modules and positional embeddings to introduce the proper inductive biases for probabilistic time-series forecasting to the model architecture.

**Summary of Contributions** In this paper, we are concerned with both improving forecast accuracy *and* reducing excess forecast volatility. We present a set of novel architectures that seek to remedy some of inductive biases that are currently missing in state of the art MQ-Forecasters (Wen et al., 2017). Our main contributions are

1. **Positional Encoding from Event Indicators**: Current MQ-Forecasters use explicitly engineered holiday "distances" to provide the model with information about the seasonality of the time series. We introduce a novel positional encoding mechanism that allows the network to learn a context-dependent seasonality function specific that is more general than conventional position encoding schemes.
2. **Horizon-Specific Decoder-Encoder Attention**: Wen et al. (2017); Madeka et al. (2018) and other MQ-Forecasters learn a single encoder representation for all future dates and periods being forecasted. We present a novel horizon-specific decoder-encoder attention scheme that allows the network to learn a representation of the past that depends on the period being forecasted.
3. **Decoder Self-Attention for Forecast Evolution**: To the best of our knowledge, this is *the first work* to consider the *impacts of network architecture design on forecast evolution*. Importantly, we accomplish this by using attention mechanisms to introduce the right inductive biases, and not by explicitly penalizing a measure of forecast variability. This allows us to maintain a single objective function without needing to make trade-offs between accuracy and volatility.

By providing MQ-Forecasters with the structure necessary to learn *context information dependent encodings*, we observe major increases in accuracy (5.5% in overall P90 quantile loss throughout the year, and **up to 33% during peak periods**) on our demand forecasting application, along with a statistically significant reduction in excess volatility (**67% reduction at P50 and 95% at P90**). We also apply MQTransformer to four public datasets, and show parity with the state-of-the-art on simple, univariate tasks. On a substantially more complex public dataset (retail forecasting) we demonstrate a **38% improvement** over the previously reported state-of-the-art, and a 5% improvement in P50 QL, 11% in P90 QL versus our baseline. Because our innovations are compatible with efficient training schemes, our architecture also achieves a significant speedup (several orders of magnitude greater throughput) over earlier transformer models for time-series forecasting.

## 2 BACKGROUND AND RELATED WORK

### 2.1 TIME SERIES FORECASTING

Formally, the task considered in our work is the high-dimensional regression problem

$$p(y_{t+1,i}, \ldots, y_{t+H,i} | \mathbf{y}_{:t,i}, \mathbf{x}_{:t,i}^{(h)}, \mathbf{x}_{:t,i}^{(f)}, \mathbf{x}_i^{(s)}), \tag{1}$$

where $y_{t+s,i}, \mathbf{y}_{:t,i}, \mathbf{x}_{:t,i}^{(h)}, \mathbf{x}_{t:,i}^{(f)}, \mathbf{x}_i^{(s)}$ denote future observations of the target time series $i$, observations of the target time series observed up until time $t$, the past covariates, known future information, and static covariates, respectively.

For sequence modeling problems, Seq2Seq (Sutskever et al., 2014) is the canonical deep learning framework and although they applied this architecture to neural machine translation (NMT) tasks, it has since been adapted to time series forecasting (Nascimento et al., 2019; Yu et al., 2017; Gasparin

et al., 2019; Mukhoty et al., 2019; Wen et al., 2017; Salinas et al., 2020; Wen and Torkkola, 2019). The MQ-Forecaster framework (Wen et al., 2017) solves (1) above by treating each series $i$ as a sample from a joint stochastic process and feeding into a neural network which predicts $Q$ quantiles for each horizon. These types of models have limited contextual information available to the decoder – a drawback inherited from the Seq2Seq architecture – as it produces each estimate $y_{t+s,i}^q$, the $q^{th}$ quantile of the distribution of the target at time $t + s$, $y_{t+s,i}$. Seq2Seq models rely on a single encoded context to produce forecasts for all horizons, imposing an information bottleneck and making it difficult for the model to understand long term dependencies.

Our MQTransformer architecture uses the direct strategy: the model outputs the quantiles of interest directly, rather than the parameters of a distribution from which samples are to be generated. This has been shown (Wen et al., 2017) to outperform parametric models, like DeepAR (Salinas et al., 2020), on a wide variety of tasks. Recently, Lim et al. (2019) consider an application of attention to multi-horizon forecasting, but their method still produces a single context for all horizons and by using an RNN decoder, does not enjoy the same scaling properties as MQ-Forecaster models.

## 2.2 ATTENTION MECHANISMS

Bahdanau et al. (2014) introduced the concept of an attention mechanism to solve the information bottleneck and sequence alignment problems in Seq2Seq architectures for NMT. Recently, attention has enjoyed success across a diverse range of applications including natural language processing (NLP), computer vision (CV) and time-series forecasting tasks (Galassi et al., 2019; Xu et al., 2015; Shun-Yao Shih and Fan-Keng Sun and Hung-yi Lee, 2019; Kim and Kang, 2019; Cinar et al., 2017; Li et al., 2019; Lim et al., 2019). Many variants have been proposed including self-attention and dot-product attention (Luong et al., 2015; Cheng et al., 2016; Vaswani et al., 2017; Devlin et al., 2019), and transformer architectures (end-to-end attention with no recurrent layers) achieve state-of-the-art performance on most NLP tasks.

Time series forecasting applications exhibit seasonal trends and the absolute position encodings commonly used in the literature cannot be applied. Our work differs from previous work on *relative position encodings* (Dai et al., 2019; Huang et al., 2018; Shaw et al., 2018) in that we learn a representation from a time series of indicator variables which encode events relevant to the target application (such as holidays and promotions). This imposes a strong inductive bias that allows the model to generalize well to future observations. Existing encoding schemes either involve feature engineering (e.g. sinusoidal encodings) or have a maximum input sequence length; ours requires no feature engineering – the model learns it directly from raw data – and it extends to arbitrarily long sequences. See Appendix A for additional background on attention mechanisms.

## 2.3 MARTINGALE DIAGNOSTICS

Originally the *martingale model of forecast evolution* (MMFE) was conceived as a way to simulate demand forecasts used in inventory planning problems (Heath and Jackson, 1994). Denoting by $\widehat{Y}_{T|t}$ the forecast for $Y_T$ made at time $t \leq T$, the MMFE assumes that the forecast process $\{\widehat{Y}_{T|t}\}_t$ is martingale. Informally, a martingale captures the notion that a forecast should use all information available to the forecasting system at time $t$. Mathematically, a discrete time martingale is a stochastic process $\{X_t\}$ such that

$$\mathbb{E}[X_{t+1}|X_t, \ldots, X_1] = X_t.$$

We assume a working knowledge of martingales and direct the reader to Williams (1991) for a thorough coverage in discrete time.

Taleb (2018) describe how martingale forecasts correspond to rational updating, then expanded by Augenblick and Rabin (2019). Taleb (2018), Taleb and Madeka (2019) and Augenblick and Rabin (2019) go on to develop tests for forecasts that rule out martingality and indicate irrational or predictable updating for binary bets. Foster and Stine (2021) further extend these ideas to quantile forecasts. They consider the coverage probability process $p_t := \mathbb{P}[Y_T \leq \tau | Y_s, s \leq t] = \mathbb{E}[I(Y_T \leq \tau) | Y_s, s \leq t]$, where $\tau$ denotes the forecast announced in the first period $t = 0$. Because $\{p_t\}$ is also a martingale, the authors show that $\mathbb{E}[(p_T - \pi)^2] = \sum_{t=1}^T \mathbb{E}[(p_t - p_{t-1})^2] = \pi(1 - \pi)$, where $\pi = p_0$ is the expected value of $p_T$, a Bernoulli random variable, across realizations of the coverage process. In the context of quantile forecasting, $\pi$ is simply the quantile forecasted.

The measure of excess volatility proposed is the quadratic variation process associated with $\{p_t\}$, $Q_s := \sum_{t=0}^{s}(p_t - p_{t-1})^2$. While this process is not a martingale, we do know that under the MMFE assumption, $\mathbb{E}[Q_T] = \pi(1 - \pi)$. The martingale $V_t := Q_t - (p_t - \pi)^2$ is obtained from this process in the usual way by subtracting the compensator. In Section 4 we leverage the properties of $\{V_t\}$ to compare the dynamics of forecasts produced by a variety of models, demonstrating that our feedback-aware decoder self-attention units reduce excess forecast volatility.

**Notation** We denote by $H$ and $Q$ the number of horizons and quantiles being forecast, respectively. Bolded characters are used to indicate vector and matrix values. The concatenation of two vectors $\mathbf{v}$ and $\mathbf{u}$ is denoted as $[\mathbf{u}; \mathbf{v}]$.

## 3 METHODOLOGY

In this section we present our MQTransformer architecture, building upon the MQ-Forecaster framework (Wen et al., 2017). We extend the MQ-Forecaster family of models because it, unlike many other architectures considered in the literature, can be applied at a large-scale (millions of samples) due to its use of *forking sequences* – a technique to dramatically increase throughput during training and avoid expensive data augmentation.

For ease of exposition, we reformulate the generic probabilistic forecasting problem in (1) as

$$p(y_{t+1,i}, \ldots, y_{t+H,i} | \mathbf{y}_{:t,i}, \mathbf{x}_{:t,i}, \mathbf{x}_i^{(l)}, \mathbf{x}^{(g)}, \mathbf{x}_i^{(s)}),$$

where $\mathbf{x}_{:t,i}$ are past observations of all covariates, $\mathbf{x}_i^{(l)} = \{\mathbf{x}_{s,i}^{(l)}\}_{s=1}^{\infty}$ are known covariates specific to time-series $i$, $\mathbf{x}^{(g)} = \{\mathbf{x}_s^{(g)}\}_{s=1}^{\infty}$ are the global, known covariates. In this setting, known signifies that the model has access to (potentially noisy) observations of past and future values. Note that this formulation is equivalent to (1), and that known covariates can be included in the past covariates $\mathbf{x}_{:t}$. When it can be inferred from context, the time series index $i$ is omitted.

**Learning Objective** We train a quantile regression model to minimize the quantile loss, summed over all forecast creation times (FCTs) and quantiles

$$\sum_t \sum_q \sum_k L_q\left(y_{t+k}, \widehat{y}_{t+k}^{(q)}\right),$$

where $L_q(y, \widehat{y}) = q(y - \widehat{y})_+ + (1-q)(\widehat{y} - y)_+$, $(\cdot)_+$ is the positive part operator, $q$ denotes a quantile, and $k$ denotes the horizon.

### 3.1 NETWORK ARCHITECTURE

The design of the architecture is similar to MQ-(C)RNN (Wen et al., 2017), and consists of encoder, decoder and position encoding blocks (see Figure 5 in Appendix B). The position encoding outputs, for each time step $t$, are a representation of global position information, $\mathbf{r}_t^{(g)} = \mathrm{PE}_t^{(g)}(\mathbf{x}_i^{(g)})$, as well as time-series specific context information, $\mathbf{r}_t^{(l)} = \mathrm{PE}_t^{(l)}(\mathbf{x}_i^{(l)})$. Intuitively, $\mathbf{r}_t^{(g)}$ captures position information that is independent of the time-series $i$ (such as holidays), whereas $\mathbf{r}_t^{(l)}$ encodes time-series specific context information (such as promotions). In both cases, the inputs are a time series of indicator variables *and require no feature-engineering or handcrafted functions*.

The encoder then summarizes past observations of the covariates into a sequence of hidden states $\mathbf{h}_t := \mathrm{encoder}(\mathbf{y}_{:t}, \mathbf{x}_{:t}, \mathbf{r}_{:t}^{(g)}, \mathbf{r}_{:t}^{(l)}, \mathbf{s})$. Using these representations, the decoder produces an $H \times Q$ matrix of forecasts $\widehat{\mathbf{Y}}_t = \mathrm{decoder}(\mathbf{h}_{:t}, \mathbf{r}^{(g)}, \mathbf{r}^{(l)})$. Note that in the decoder, the model has access to position encodings.

In this section we focus on our novel attention blocks and position encodings; the reader is directed to Appendix B for additional architecture details.

**MQTransformer** Following the generic pattern given above, we present the MQTranformer architecture. First, define the combined position encoding as $\mathbf{r} := [\mathbf{r}^{(g)}; \mathbf{r}^{(l)}]$. In the encoder we use a stack of dilated temporal convolutions (van den Oord et al., 2016; Wen et al., 2017) to encode historical time-series and a multi-layer perceptron to encode the static features as (2).

Table 1: MQTransformer encoder and decoder

| ENCODER | | DECODER CONTEXTS | |
|---|---|---|---|
| $\mathbf{h}_t^1 = \text{TEMPORALCONV}(\mathbf{y}_{:t}, \mathbf{x}_{:t}, \mathbf{r}_{:t})$ | (2) | $\mathbf{c}_{t,h} = \text{HSATTENTION}(\mathbf{h}_{:t}, \mathbf{r})$ | (3) |
| $\mathbf{h}_t^2 = \text{FEEDFORWARD}(\mathbf{s})$ | | $\mathbf{c}_t^a = \text{FEEDFORWARD}(\mathbf{h}_t, \mathbf{r})$ | |
| $\mathbf{h}_t = [\mathbf{h}_t^1; \mathbf{h}_t^2],$ | | $\mathbf{c}_t = [\mathbf{c}_{t,1}; \cdots ; \mathbf{c}_{t,H}; \mathbf{c}_t^a]$ | |
| | | $\widetilde{\mathbf{c}}_{t,h} = \text{DSATTENTION}(\mathbf{c}_{:t}, \mathbf{h}_{:t}, \mathbf{r}),$ | |

Our decoder incorporates our horizon specific and decoder self-attention blocks, and consists of two branches. The first (global) branch summarizes the encoded representations into horizon-specific ($\mathbf{c}_{t,h}$) and horizon agnostic ($\mathbf{c}_t^a$) contexts. Formally, the global branch $\mathbf{c}_t := m_G(\cdot)$ is given by (3).

The output branch consists of a self-attention block followed by a local MLP, which produces outputs using the same weights for each horizon. For FCT $t$ and horizon $h$, the output is given by $(\widehat{y}_{t+h}^1, \ldots, \widehat{y}_{t+h}^Q) = m_L(\mathbf{c}_t^a, \mathbf{c}_{t,h}, \widetilde{\mathbf{c}}_{t,h}, \mathbf{r}_{t+h})$. Next we describe the specifics of our position encoding and attention blocks.

## 3.2 LEARNING POSITION AND CONTEXT REPRESENTATIONS FROM EVENT INDICATORS

Prior work typically uses a variant on one of two approaches to provide attention blocks with position information: (1) a handcrafted representation (such as sinusoidal encodings) or (2) a matrix $\mathbf{M} \in \mathbb{R}^{L \times d}$ of position encoding where $L$ is the maximum sequence length and each row corresponds to the position encoding for time point.

In contrast, our novel encoding scheme maps sequences of indicator variables to a $d$-dimensional representations. For demand forecasting, this enables our model to learn an arbitrary function of events (like holidays and promotions) to encode position information. As noted above, our model includes two position encodings: $r_t^{(g)} := PE_t^{(g)}(\mathbf{x}^{(g)})$ and $r_t^{(l)} := PE_t^{(l)}(\mathbf{x}^{(l)})$, one that is shared among all time-series $i$ and one that is specific. For the design we use in Section 4, $PE^{(g)}$ is implemented as a bidirectional (looking both forward and backward in time) 1-D convolution and $PE^{(l)}$ is an MLP applied separately at each time step. For reference, MQ-(C)RNN (Wen et al., 2017) uses linear holiday and promotion distances to represent position information.

By using 1-D convolutions and MLPs over a time-series of indicator variables, our approach is more flexible and can be used to learn a position representation that is context specific. In the demand forecasting application, consider two products during a holiday season where one has a promotion and the other does not. These two products need different position encodings, else the decoder-encoder attention (Section 3.3) will not be able to align target horizons with past contexts. In addition, note that the classical method of learning a matrix embedding $\mathbf{M}$ can be recovered as a special case of our approach. Consider a sequence of length $L$, and take $\mathbf{x}^{(g)} := [\mathbf{e}_1, \ldots, \mathbf{e}_L]$, where $\mathbf{e}_s$ is used to denote the vector in $\mathbb{R}^L$ with a 1 in the $s^{th}$ position and 0s elsewhere. To recover the matrix embedding scheme, we define $\text{PE}_t^{\text{matrix}}(\mathbf{x}^{(g)}) := \mathbf{x}_t^{(g), \top} \mathbf{M}$.

## 3.3 CONTEXT DEPENDENT AND FEEDBACK-AWARE ATTENTION

### HORIZON-SPECIFIC DECODER-ENCODER ATTENTION

To motivate the horizon-specific attention unit, consider a retail demand forecasting task. At each FCT $T$, the forecaster produces forecasts for multiple target horizons, which may contain different events such as promotions or holidays. Prior work (Wen et al., 2017) incorporated horizon-specific contexts with the functional form $c_{t,h} = f(h_t, r_{t+h})$ for each FCT, target horizon. Because events like promotions or holidays – which are strong predictors of observed demand – are fairly sparse, this functional form is insufficient. Instead, a function of the form $c_{t,h} = f(h_1, r_1, \ldots, h_t, r_t, r_{t+h})$ allows the model to use information from many past periods, which is valuable due to the sparsity of relevant events. Using an attention mechanism to align target horizons enables this. Figure 1 depicts the difference between prior work and the horion-specific attention in MQTransformer.

Table 2: Attention weight and output computations for blocks introduced in Section 3.3

| BLOCK | ATTENTION WEIGHTS | | OUTPUT | |
|---|---|---|---|---|
| DECODER-ENCODER ATTENTION | $A^h_{t,s} = \mathbf{q}^{h,\top}_t \mathbf{W}^\top_q \mathbf{W}_k \mathbf{k}_s$ $\mathbf{q}^h_t = [\mathbf{h}_t; \mathbf{r}_t; \mathbf{r}_{t+h}]$ $\mathbf{k}_s = [\mathbf{h}_s; \mathbf{r}_s]$ $\mathbf{v}_s = \mathbf{h}_s$ | (4) | $\mathbf{c}_{t,h} = \sum_{s=t-L}^{t} A^h_{t,s} \mathbf{W}_v \mathbf{v}_s$ | (5) |
| DECODER SELF-ATTENTION | $A^h_{t,s,r} = \mathbf{q}^\top_{t,h} \mathbf{W}^{h,\top}_q \mathbf{W}^h_k \mathbf{k}_{s,r}$ $\mathbf{q}_{t,h} = [\mathbf{h}_t; \mathbf{c}_{t,h}; \mathbf{r}_t; \mathbf{r}_{t+h}]$ $\mathbf{k}_{s,r} = [\mathbf{c}_{s,r}; \mathbf{r}_s; \mathbf{r}_{s+r}]$ $\mathbf{v}_{s,r} = \mathbf{c}_{s,r}$ | (6) | $\widetilde{\mathbf{c}}_{t,h} = \sum_{(s,r)\in\mathcal{H}(t,h)} A^h_{s,t,r} \mathbf{W}^h_v \mathbf{v}_{s,r},$ $\mathcal{H}(t,h) := \{(s,r)|s+r=t+h\}$ | (7) |

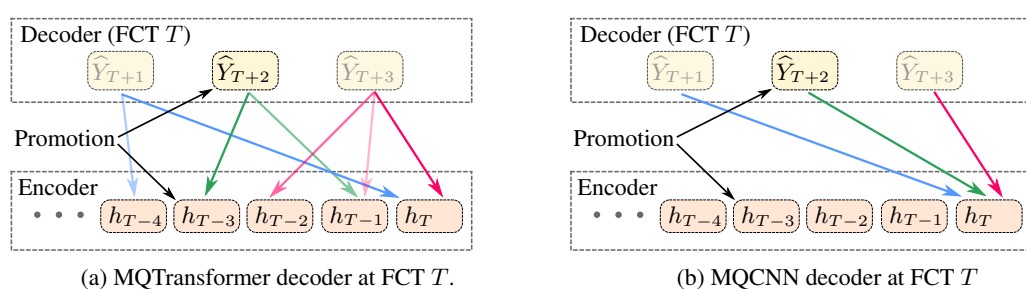

(a) MQTransformer decoder at FCT $T$.          (b) MQCNN decoder at FCT $T$

Figure 1: Example of a demand forecasting task where periods $T-3$ and $T+2$ both have promotions. The encoded context $h_{T-3}$, the last time the item had a promotion, contains useful information for forecasting for target periods that also have a promotion. The horizon-specific attention aligns past encoded contexts with the target horizon.

To do this, we introduce the horizon-specific attention mechanism, which can be viewed is a multi-headed attention mechanism where the projection weights are shared across all horizons. Each head corresponds to a different horizon. It differs from a traditional multi-headed attention mechanism in that its purpose is to attend over representations of past time points to produce a representation specific to the target period. In our architecture, the inputs to the block are the encoder hidden states and position encodings. Mathematically, for time $s$ and horizon $h$, the attention weight for the value at time $t$ is computed as (4).

Observe that there are two key differences between these attention scores and those in the vanilla transformer architecture: (a) projection weights are shared by all $H$ heads, (b) the addition of the position encoding of the target horizon $h$ to the query. The output of our horizon specific decoder-encoder attention block, $\mathbf{c}_{t,h}$, is obtained by taking a weighted sum of the encoder hidden contexts, up to a maximum look-back of $L$ periods as in (5).

DECODER SELF-ATTENTION

The martingale diagnostic tools developed in (Foster and Stine, 2021) indicate a deep connection between accuracy and volatility. We leverage this connection to develop a novel decoder self-attention scheme for multi-horizon forecasting. To motivate the development, consider a model which forecasts values of 40, 60 when the demand has constantly been 50 units. We would consider this model to have excess volatility. Similarly, a model forecasting 40, 60 when demand jumps between 40 and 60 units would not be considered to have excess volatility. This is because the first model fails to learn from its past forecasts - it continues jumping between 40, 60 when the demand is 50 units.

In order to ameliorate this, we need to pass information of the previous forecast errors into the current forecast. For each FCT $t$ and horizon $h$, the model attends on the previous forecasts using a query containing the demand information for that period. The attention mechanism has a separate head for each horizon.

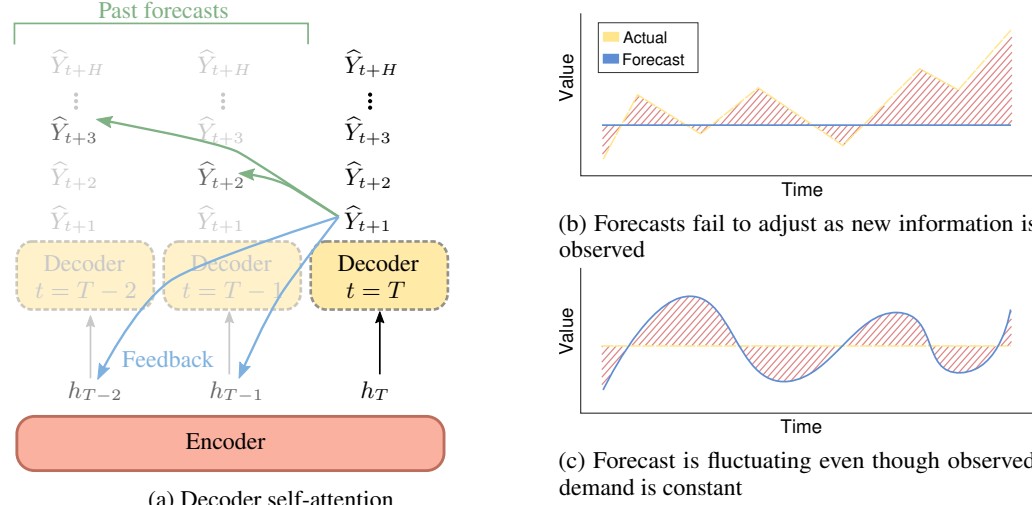

(a) Decoder self-attention

(b) Forecasts fail to adjust as new information is observed

(c) Forecast is fluctuating even though observed demand is constant

Figure 2: The decoder attends over past forecasts for the *same target horizon* – the context $h_t$ contains feedback information (demand and other encoded signals), allowing the model to adjust forecasts that are either too volatile or not volatile enough as the target date approaches.

Rather than attend on the demand information and prior outputs directly, a richer representation of the same information is used: the demand information at time $t$ is incorporated via the encoded context $\mathbf{h}_t$ and previous forecasts are represented via the corresponding horizon-specific context $\mathbf{c}_{s,r}$ – in the absence of decoder-self attention $\mathbf{c}_{s,r}$ would be passed through the local MLP to generate the forecasts. Formally, the attention scores are given by (6). The horizon-specific and feedback-aware outputs, $\widetilde{\mathbf{c}}_{t,h}$, are given by (7). Note how we sum only over previous forecasts of the same period. In Section 4 we demonstrate the importance of this inductive bias by ablating against a more traditional decoder-self attention unit that attends over all past forecasts.

## 4 EMPIRICAL RESULTS

### 4.1 LARGE-SCALE DEMAND FORECASTING

First, we evaluate our architecture on a demand forecasting problem for a large-scale e-commerce retailer with the objective of producing multi-horizon forecasts for the next 52 weeks. We conduct our experiments on a subset of products ($\sim$ 2 million products) in the US store. Each model is trained using a single machine with 8 NVIDIA V100 Tensor Core GPUs, on three years of demand data (2015-2018); one year (2018-2019) is held out for back-testing.

To assess the effects of each innovation, we ablate by removing components one at a time. The architectures we compare are the baseline (MQ-CNN), MQTransformer (MQT), MQTransformer without decoder self-attention (MQT-NoDS), and MQ-Transformer with a decoder self-attention unit that attends over all past forecasts (MQT-All). MQ-CNN is selected as the baseline since prior work[2] demonstrate that MQ-CNN outperforms MQ-RNN and DeepAR on this dataset, and as can be seen in Table 4, MQ-CNN similarly outperforms MQ-RNN and DeepAR on the public retail forecasting dataset. For additional details see Appendix C (Table 5 describes the decoder-self attention unit used in MQT-All).

**Forecast Accuracy** Table 3 summarizes several key metrics that demonstrate the accuracy improvements achieved by adding our proposed attention mechanisms to the MQ-CNN architecture. We consider overall quantile loss, as well as quantile loss for specific target periods (seasonal peaks, promotions). We also measure post-peak ramp-down performance – models that suffer issues with alignment will continue to forecast high for target weeks after a seasonal peak. By including MQ-

---

[2]Wen et al. (2017), Figure 3 shows MQ-CNN (labeled "MQ_CNN_wave") outperforms MQ-RNN (all variants) and DeepAR (labeled "Seq2SeqC") on the test set.

Table 3: P50 (50th percentile) and P90 (90th percentile) quantile loss on the backtest year along different dimensions. Values indicate relative performance versus the baseline.

| | MQ-CNN | | MQT | | MQT-NoDS | | MQT-ALL | |
| DIMENSION | P50 | P90 | P50 | P90 | P50 | P90 | P50 | P90 |
|---|---|---|---|---|---|---|---|---|
| OVERALL | 1.000 | 1.000 | **0.977** | **0.945** | 0.983 | 0.963 | **0.977** | 0.957 |
| SEASONAL PEAK 1 | 1.000 | 1.000 | **0.865** | **0.667** | 0.881 | 0.688 | 0.908 | 0.729 |
| SEASONAL PEAK 2 | 1.000 | 1.000 | 0.957 | **0.884** | 0.984 | 0.946 | **0.953** | 0.931 |
| SEASONAL PEAK 3 | 1.000 | 1.000 | 0.838 | **0.845** | **0.837** | 0.892 | 0.851 | 0.870 |
| POST-PEAK RAMPDOWN | 1.000 | 1.000 | 0.987 | **0.978** | **0.985** | 0.997 | 0.990 | 0.991 |
| PROMOTION TYPE 1 | 1.000 | 1.000 | **0.945** | **0.800** | 0.969 | 0.824 | 0.953 | 0.842 |
| PROMOTION TYPE 2 | 1.000 | 1.000 | **0.868** | **0.751** | 0.895 | 0.762 | 0.914 | 0.808 |
| PROMOTION TYPE 3 | 1.000 | 1.000 | **0.949** | **0.837** | 1.075 | 1.004 | 0.960 | 0.911 |

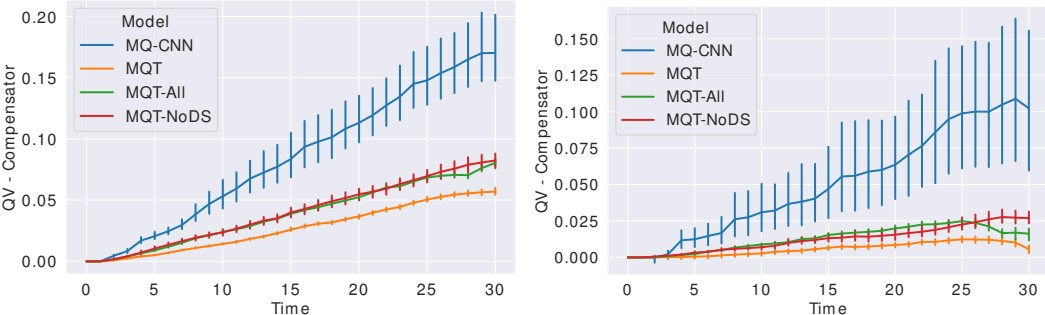

Figure 3: Martingale diagnostic process $\{V_t\}$ for P50 (left) and P90 (right) forecasts. Trajectories are demand-weighted and the results are averaged over all items, target weeks in the test period (2018-2019). Closer to zero is better.

Transformer's attention mechanisms in the architecture, we see up to 33% improvements for seasonal peaks and 24.9% improvements on promotions versus MQ-CNN. Further, the ablation analysis (against MQT-NoDS and MQT-All) shows that both novel attention units are important to improving accuracy, and that our decoder-self attention scheme (where we attend only over past forecasts for the same target period) introduces a powerful inductive bias. Table 8 in Appendix C shows the metrics computed on only short horizons, where we see even greater gains in accuracy.

**Forecast Volatility** We study the effect of our proposed attention mechanisms on excess forecast volatility using diagnostic tools recently proposed by Foster and Stine (2021). Coverage probabilities are computed by fitting a gamma distribution to the two quantiles produced for each FCT, target horizon. Figure 3 shows $\{V_t\}$ (see Section 2) for the 4 models we consider, along with bootstrapped 95% confidence intervals. The results are weighted by demand – in the context of bull-whip effects in a supply chain, excess volatility will have the largest impact for items with high demand – but similar results are obtained if we weight all items equally. Under the MMFE, the lines should appear horizontal and any deviation above this (on an aggregate level) indicates excess volatility in the forecast evolution. While none of the models produce ideal forecasts, MQT has the least amount of excess volatility – it shows statistically significant reductions over all other models – and achieves a 67% reduction over baseline at P50 and 95% at P90.

## 4.2 PUBLICLY AVAILABLE DATASETS

Following Lim et al. (2019), we consider applications to brick-and-mortar retail sales, electricity load, securities volatility and traffic forecasting. In each task, we report the quantile loss summed over all forecast creation times, normalized by the target values:

$$\frac{2 \sum_t \sum_k L_q \left( y_{t+k}, \widehat{y}_{t+k}^{(q)} \right)}{\sum_t \sum_k |y_{t+k}|}.$$

Table 4: Quantile loss metrics with the best results on each task emphasized. Results in parentheses correspond to training MQTransformer without forking sequences on 450K trajectories only.

|  | TASK | DEEPAR | CONVTRANS | MQ-RNN | MQ-CNN | TFT | MQTRANSFORMER |
|---|---|---|---|---|---|---|---|
| **P50 QL** | ELECTRICITY | 0.075 | 0.059 | 0.077 | 0.076 | **0.055** | 0.057 |
|  | RETAIL | 0.574 | 0.429 | 0.379 | 0.269 | 0.354 | **0.256** (0.2645) |
|  | VOLATILITY | 0.050 | 0.047 | 0.042 | 0.042 | **0.039** | **0.039** |
|  | TRAFFIC | 0.161 | 0.122 | 0.117 | 0.115 | **0.095** | 0.101 |
|  | TASK | DEEPAR | CONVTRANS | MQ-RNN | MQ-CNN | TFT | MQTRANSFORMER |
| **P90 QL** | ELECTRICITY | 0.040 | 0.034 | 0.036 | 0.035 | **0.027** | **0.027** |
|  | RETAIL | 0.230 | 0.192 | 0.152 | 0.118 | 0.147 | **0.106** (0.109) |
|  | VOLATILITY | 0.024 | 0.024 | 0.021 | 0.020 | 0.020 | **0.019** |
|  | TRAFFIC | 0.099 | 0.081 | 0.082 | 0.077 | 0.070 | **0.068** |

For the retail task, we predict the next 30 days of sales, given the previous 90 days of history. This dataset contains a rich set of static, time series, and known features. At the other end of the spectrum, the electricity load dataset is univariate. See Appendix D for additional information about these tasks. Table 4 compares MQTransformer's performance with other recent works[3] – DeepAR (Salinas et al., 2020), ConvTrans (Li et al., 2019), MQ-RNN (Wen et al., 2017), and TFT (Lim et al., 2019).

Our MQTransformer architecture is competitive with or beats the state-of-the-art on the electricity load, volatility and traffic prediction tasks, as shown in Table 4. On the most challenging task, it dramatically outperforms the previously reported state of the art by 38% and the MQ-CNN baseline by 5% at P50 and 11% at P90. Because MQ-CNN and MQTransformer are trained using forking sequences, we can use the entire training population, rather than downsample as is required to train TFT (Lim et al., 2019). To ascertain what portion of the gain is due to learning from more trajectories, versus our innovations alone, we retrain the optimal MQTransformer architecture using a random sub-sample of 450K trajectories (the same sampling procedure as TFT) and without using forking sequences – the results are indicated in parentheses in Table 4. We can observe that MQTransformer still dramatically outperforms TFT, and its performance is similar to the MQ-CNN baseline trained on all trajectories. See Appendix D for more details and results on the Favorita forecasting task.

**Computational Efficiency** For purposes of illustration, consider the Retail dataset. Prior work (Lim et al., 2019) was only able to make use of 450K out of 20M trajectories, and the optimal TFT architecture required 13 minutes per epoch (minimum validation error at epoch 6)[4] using a single V100 GPU. Our innovations are compatible with forking sequences, and thus our architecture can make use of *all available trajectories*. To train, MQTransformer requires only 5 minutes per epoch (on 20M trajectories) using a single V100 GPU (minimum validation error reached after 5 epochs).Some of the differences in runtime can be attributed to use of different deep learning frameworks, but it is clear that MQTransformer can be trained much more efficiently than models like TFT and DeepAR.

## 5 CONCLUSIONS AND FUTURE WORK

In this work, we present three novel architecture enhancements that improve bottlenecks in state of the art MQ-Forecasters. To the best of our knowledge, this is the first work to consider the impact of model architecture on forecast evolution. On a large scale demand-forecasting task we demonstrated how position embeddings can be learned directly from domain-specific event indicators and horizon-specific contexts can improve performance for difficult sub-problems such as promotions or seasonal peaks. Together, these innovations produced significant improvements in accuracy across different dimensions and in the excess variation of the forecast – we also saw there was no tradeoff between these two desiderata, but rather they are closely linked. On a public retail forecasting task MQTransformer outperformed the baseline architecture by 5% at P50, 11% at P90 and the previous reported state-of-the-art (TFT) by 38%. Beyond accuracy gains, our model achieves massive increases in throughput compared to existing transformer architectures for time-series forecasting.

---

[3]Results for TFT, MQ-RNN, DeepAR and ConvTrans are from Lim et al. (2019).

[4]Timing results obtained by running the source code provided by Lim et al. (2019)

An interesting direction we intend to explore in future work is incorporating encoder self-attention so that the model can leverage arbitrarily long historical time series, rather than the fixed length consumed by the convolution encoder.

REPRODUCIBILITY STATEMENT

Our work included experiments on both public and proprietary datasets. Although, by definition, the experiments on the proprietary dataset cannot be reproduced by others, we did perform experiments using a public dataset with very similar features (Favorita forecasting task). In terms of reproducing our results on the public datasets, we used publicly available code to preprocess the data and construct model inputs (see Footnote 3). Since we were unable to obtain permission to release the source code for our model architecture, we instead included sufficient detail such that it will be straightforward for others to re-implement. See Table 1, Table 2 and Appendix B for details about the exact structure of the MQTransformer architecture. See Table 9 and Appendix D for details about the hyperparameters used for each of the public datasets.

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

# A ADDITIONAL BACKGROUND AND RELATED WORK

## A.1 ATTENTION MECHANISMS

Attention mechanisms can be viewed as a form of content based addressing, that computes an alignment between a set of *queries* and *keys* to extract a *value*. Formally, let $\mathbf{q}_1, \ldots, \mathbf{q}_t, \mathbf{k}_1, \ldots, \mathbf{k}_t$ and $\mathbf{v}_1, \ldots, \mathbf{v}_t$ be a series of queries, keys and values, respectively. The $s^{th}$ attended value is defined as $\mathbf{c}_s = \sum_{i=1}^{t} \text{score}(\mathbf{q}_s, \mathbf{k}_t)\mathbf{v}_t$, where score is a scoring function – commonly $\text{score}(\mathbf{u}, \mathbf{v}) := \mathbf{u}^\top \mathbf{v}$. In the vanilla transformer model, $\mathbf{q}_s = \mathbf{k}_s = \mathbf{v}_s = \mathbf{h}_s$, where $\mathbf{h}_s$ is the hidden state at time $s$. Because attention mechanisms have no concept of absolute or relative position, some sort of position information must be provided. Vaswani et al. (2017) uses a sinusoidal positional encoding added to the input to an attention block, providing each token's position in the input time series.

In the vanilla transformer (Vaswani et al., 2017), a sinusoidal position embedding is added to the network input and each encoder layer consists of a multi-headed attention block followed by a feed-forward sub-layer. For each head $i$, the attention score between query $q_s$ and key $k_t$ is defined as follows for the input layer

$$A_{s,t}^h = (\mathbf{x}_s + \mathbf{r}_s)^\top \mathbf{W}_q^{h,\top} \mathbf{W}_k^h (\mathbf{x}_t + \mathbf{r}_t) \tag{8}$$

where $\mathbf{x}_s, \mathbf{r}_s$ are the observation of the time series and the position encoding, respectively, at time $s$. In Section 3 we introduced attention mechanisms that differ in their treatment of the position dependent biases

# B MQTRANSFORMER ARCHITECTURE DETAILS

In this section we describe in detail the layers in our MQTransformer architecture, which is based off of the MQ-Forecaster framework (Wen et al., 2017) and uses a wavenet encoder (van den Oord et al., 2016) for time-series covariates. Before describing the layers in each component, Figure 5 outlines the MQTransformer architecture. On different datasets, we consider the following variations: choice of encoding for categorical variables, a tunable parameter $d_h$ (dimension of hidden layers), dropout rate $p_{drop}$, a list of dilation rates for the wavenet encoder, and a list of dilation rates for the position encoding. The ReLU activation function is used throughout the network.

## B.1 INPUT EMBEDDINGS AND POSITION ENCODING

Static categorical variables are encoded using either one-hot encoding or an embedding layer. Time-series categorical variables are one-hot encoded, and then passed through a single feed-forward layer of dimension $d_h$.

The global position encoding module takes as input the known time-series covariates, and consist of a stack of dilated, bi-directional 1-D convolution layers with $d_h$ filters. After each convolution is a ReLU activation, followed by a dropout layer with rate $p_{drop}$, and the local position encoding is implemented as a single dense layer of dimension $d_h$. Figure 4b shows an example of the position encoding from a model trained on the large-scale demand forecasting task.

## B.2 ENCODER

After categorical encodings are applied, the inputs are passed through the encoder block. The encoder consists of two components: a single dense layer to encode the static features, and a stack of dilated, temporal convolutions. The time-series covariates are concatenated with the position encoding to form the input to the convolution stack. The output of the encoder block is produced by replicating the encoded static features across all time steps and concatenating with the output of the convolution.

## B.3 DECODER

Please see Table 1 for a description of the blocks in the decoder. The dimension of each head in both the horizon-specific and decoder self-attention blocks is $d_h/2$. The dense layer used to compute $\mathbf{c}_t^a$ has dimension $d_h/2$. The output block is two layer MLP with hidden layer dimension $d_h/2$,

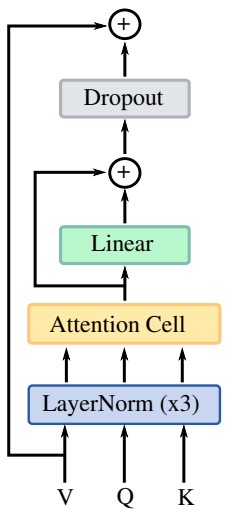

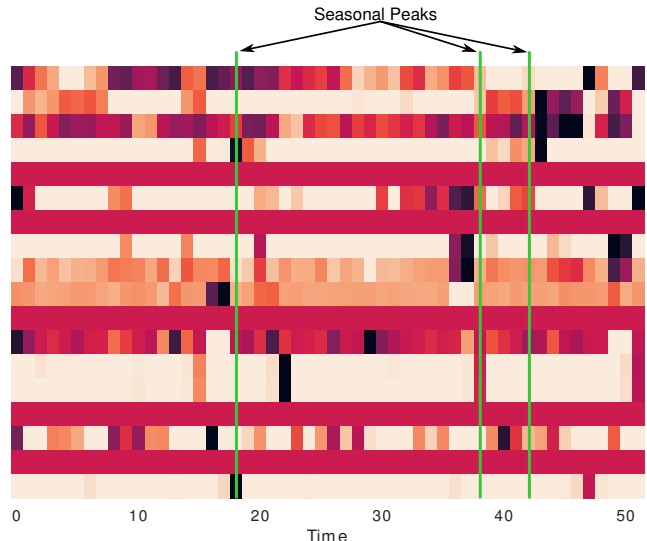

(a) Basic structure of our time-series attention blocks

(b) Position encoding learned from event indicators in the large scale demand forecasting task (Section 4)

Figure 4: Components of the MQTransformer architecture

and weights are shared across all time points and horizons. The output layer has one output per horizon, quantile pair. Figure 4a shows the structure of both our attention blocks, where the sub-layer "Attention Cell" is replaced with either the decoder-encoder or decoder-self attention.

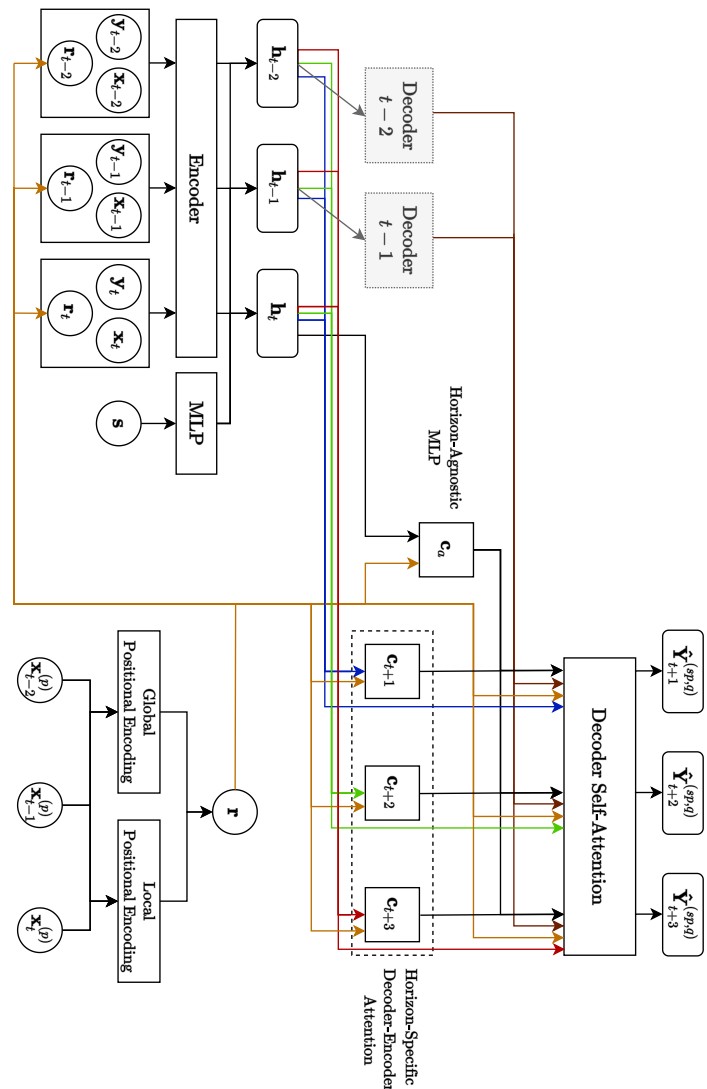

Figure 5: MQTransformer architecture with learned global/local positional encoding, horizon-specific decoder-encoder attention, and decoder self-attention

## C   LARGE SCALE DEMAND FORECASTING EXPERIMENTS

### C.1   ABLATION STUDY

In the ablation study, the model MQT-All uses a standard multi-head attention applied to the concatenated contexts for all horizons at each period $t$. A separate head is used for each output horizon (52 heads on the private dataset).

Table 5: Attention weight and output computations for MQT-All decoder self-attention

| ATTENTION WEIGHTS | OUTPUT |
|---|---|
| $A_{t,s}^h = \mathbf{q}_t^\top \mathbf{W}_q^{h,\top} \mathbf{W}_k^h \mathbf{k}_s$ $\mathbf{q}_t = [\mathbf{c}_{s,1}; \cdots ; \mathbf{c}_{s,H}; \mathbf{h}_t; \mathbf{r}_t]$ $\mathbf{k}_s = [\mathbf{c}_{s,1}; \cdots ; \mathbf{c}_{s,H}; \mathbf{r}_s]$ $\mathbf{v}_s = [\mathbf{c}_{s,1}; \cdots ; \mathbf{c}_{s,H}]$ | $\widetilde{\mathbf{c}}_{t,h} = \displaystyle\sum_{s=t-L}^{t} A_{t,s}^h \mathbf{W}_v^h \mathbf{v}_s$ |

Table 6 gives the number of parameters in each trained model.

Table 6: Parameter counts in trained models for the large scale demand-forecasting task.

| MODEL | NUMBER OF PARAMETERS |
|---|---|
| MQ-CNN | $9.17 \times 10^5$ |
| MQT | $9.25 \times 10^5$ |
| MQT-NoDS | $9.09 \times 10^5$ |
| MQT-ALL | $2.82 \times 10^6$ |

### C.2   EXPERIMENT SETUP

In this section we describe the details of the model architecture and training procedure used in the experiments on the large-scale demand forecasting application.

#### TRAINING PROCEDURE

Because we did not have enough history available to set aside a true holdout set, all models are trained for 100 epochs, and the final model is evaluated on the test set. For the same reason, no hyperparameter tuning was performed.

#### ARCHITECTURE AND HYPERPARAMETERS

The categorical variables consist of static features of the item, and the timeseries categorical variables are event indicators (e.g. holidays). The parameters are summarized in Table 7.

### C.3   RESULTS FOR SHORTER HORIZONS

Table 8 shows results along various dimensions for horizons up to 6 weeks. Similar to the results in Section 4, MQTransformer significantly outperforms the baseline.

## D   EXPERIMENTS ON PUBLIC DATASETS

In this section we describe the experiment setup used for the public datasets in Section 4. As mentioned in Section 4.2, we display the baseline results published in Lim et al. (2019). For MQ-CNN and MQTransformer, we use their pre-processing and evaluation code to ensure parity.

Table 7: Parameter settings for Large Scale Demand Forecasting Experiments

| Parameter | Value |
|---|---|
| Encoder Convolution Dilation Rates | [1,2,4,8,16,32] |
| Position Encoding Dilation Rates | [1,2,4,8,16,20] |
| Static Categorical | One-Hot |
| Time-Series Categorical | One-Hot |
| Static Encoder Dimension | 64 |
| Convolution Filters | 32 |
| Attention Block Head Dimension | 16 |
| Dropout Rate | 0.15 |
| Activation Function | ReLU |

Table 8: Quantile loss on the backtest year along different dimensions. Values are relative performance versus baseline, with the best result for each dimension emphasized.

| | MQ-CNN | | MQT | | MQT-NoDS | | MQT-All | |
|---|---|---|---|---|---|---|---|---|
| DIMENSION | P50 | P90 | P50 | P90 | P50 | P90 | P50 | P90 |
| OVERALL | 1.000 | 1.000 | **0.962** | **0.915** | 0.983 | 0.963 | 0.977 | 0.957 |
| SEASONAL PEAK 1 | 1.000 | 1.000 | **0.865** | **0.667** | 0.881 | 0.688 | 0.908 | 0.729 |
| SEASONAL PEAK 2 | 1.000 | 1.000 | **0.925** | **0.850** | 0.972 | 0.910 | 0.932 | 0.886 |
| SEASONAL PEAK 3 | 1.000 | 1.000 | **0.798** | **0.804** | 0.803 | 0.872 | 0.815 | 0.830 |
| POST-PEAK RAMPDOWN | 1.000 | 1.000 | 0.968 | **0.956** | 0.983 | 0.996 | **0.967** | 0.975 |
| PROMOTION TYPE 1 | 1.000 | 1.000 | **0.915** | **0.601** | 0.929 | 0.616 | 0.921 | 0.628 |
| PROMOTION TYPE 2 | 1.000 | 1.000 | **0.764** | **0.537** | 0.805 | 0.561 | 0.861 | 0.640 |
| PROMOTION TYPE 3 | 1.000 | 1.000 | **0.901** | **0.746** | 1.075 | 0.942 | 0.941 | 0.848 |

## D.1 DATASETS

We evaluate our MQTransformer on four public datasets. We summarize the datasets and preprocessing logic below; the reader is referred to Lim et al. (2019) for more details. Lim et al. (2019) released their code under the Apache 2.0 License. Favorita's terms of use for their retail dataset allows for it to be used for scientific research. The Electricity and Traffic datasets are provided by the UCI machine learning repository (Dua and Graff, 2017). The volatility dataset is sourced from the Oxford-Man Institute's realized library v0.3 (Gerd Heber and Sheppard, 2009).

### RETAIL

This dataset is provided by the Favorita Corporacion (a major Grocery chain in Ecuador) as part of a Kaggle[5] to predict sales for thousands of items at multiple brick-and-mortar locations. In total there are 135K items (item, store combinations are treated as distinct entities), and the dataset contains a variety of features including: local, regional and national holidays; static features about each item; total sales volume at each location. The task is to predict log-sales for each (item, store) combination over the next 30 days, using the previous 90 days of history. The training period is January 1, 2015 through December 1, 2015. The following 30 days are used as a validation set, and the 30 days after that as the test set. These 30 day windows correspond to a single FCT. While Lim et al. (2019) extract only 450K samples from the histories during the train window, there are in fact 20M trajectories avalaible for training – because our models can produce forecasts for multiple trajectories (FCDs) simultaneously, we train using all available data from the training window.

For the volatility analysis presented in Figure 6, we used a 60 day validation window (March 1, 2016 through May 1, 2016), which corresponds to 30 FCTs.

### ELECTRICITY

This dataset consists of time series for 370 customers of at an hourly grain. The univariate data is augmented with a day-of-week, hour-of-day and offset from a fixed time point. The task is to predict hourly load over the next 24 hours for each customer, given the past seven days of usage. From the training period (January 1, 2014 through September, 1 2019) 500K samples are extracted.

### TRAFFIC

This dataset consists of lane occupancy information for 440 San Francisco area freeways. The data is aggregated to an hourly grain, and the task is to predict the hourly occupancy over the next 24 hours given the past seven days. The training period consist of all data before 2008-06-15, with the final 7 days used as a validation set. The 7 days immediately following the training window is used for evaluation. The model takes as input lane occupancy, hour of day, day of week, hours from start and an entity identifier.

### VOLATILITY

The volatility dataset consists of 5 minute sub-sampled realized volatility measurements from 2000-01-03 to 2019-06-28. Using the past one year's worth of daily measurements, the goal is to predict the next week's (5 business days) volatility. The period ending on 2015-12-31 is used as the training set, 2016-2017 as the validation set, and 2018-01-01 through 2019-06-28 as the evaluation set. The region identifier is provided as a static covariate, along with time-varying covariates daily returns, day-of-week, week-of-year and month. A log transformation is applied to the target.

## D.2 TRAINING PROCEDURE

We only consider tuning two hyper-parameters, size of hidden layer $d_h \in \{32, 64, 128\}$ and learning rate $\alpha \in \{1 \times 10^{-2}, 1 \times 10^{-3}, 1 \times 10^{-4}\}$. The model is trained using the ADAM optimizer Kingma and Ba (2015) with parameters $\beta_1 = 0.9$, $\beta_2 = 0.999$, $\epsilon = 1e - 8$ and a minibatch size of 256, for a maximum of 100 epochs and an early stopping patience of 5 epochs.

---

[5]The original competition can be found here.

We train a model for each hyperparameter setting in the search grid (6 combinations), select the one with the minimal validation loss and report the selected model's test-set error in Table 4.

### D.3 ARCHITECTURE DETAILS

Our MQTransformer architecture used for these experiments contain a single tune-able hyperparameter – hidden layer dimension $d_h$. Dataset specific settings are used for the dilation rates. For static categorical covariates we use an embedding layer with dimension $d_h$ and use one-hot encoding for time-series covariates. A dropout rate of 0.15 and ReLU activations are used throughout the network. The only difference between this variant and the one used for the non-public large scale demand forecasting task is the use of an embedding layer for static, categorical covariates rather than one-hot encoding.

### D.4 REPORTED MODEL PARAMETERS

The optimal parameters for each task are given in Table 9.

Table 9: Parameter settings of reported MQTransformer model on each public dataset.

| Name | $d_h$ | $\alpha$ | Enc. Dilation Rates | Pos. Dilation Rates |
|---|---|---|---|---|
| Electricity | 128 | $1 \times 10^{-3}$ | [1,2,4,8,16,32] | [1,2,4,8,8] |
| Traffic | 64 | $1 \times 10^{-3}$ | [1,2,4,8,16,32] | [1,2,4,8,8] |
| Volatility | 128 | $1 \times 10^{-3}$ | [1,2,4,8,16,32,64] | [1,1,2] |
| Retail | 64 | $1 \times 10^{-4}$ | [1,2,4,8,16,32] | [1,2,4,8,14] |

### D.5 MARTINGALE DIAGNOSTICS ON FAVORITA FORECASTING TASK

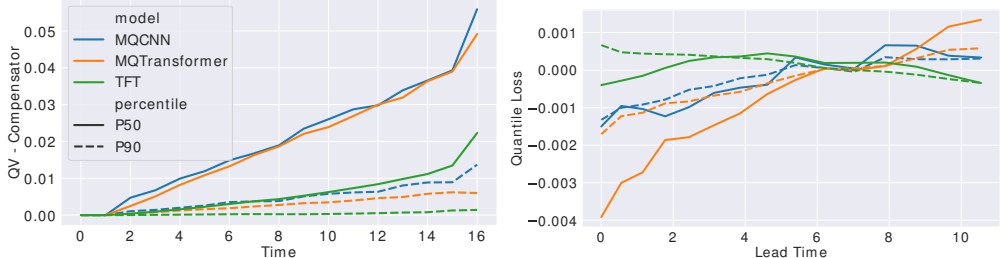

Figure 6: Forecast evolution analysis on the retail dataset. Left: Martingale Diagnostic Process $\{V_t\}$. Right: QL by lead time, averaged over target dates from 2016-03-01 through 2016-05-01; QL trajectories are centered around 0.

On the Favorita retail forecasting task, Figure 6 shows that as expected, MQTransformer substantially reduces excess volatility in the forecast evolution compared to the MQ-CNN baseline. Somewhat surprisingly, TFT exhibits much lower volatility than does MQTransformer. In Figure 6, the right hand plot displays quantile loss as the target date approaches – trajectories for each model are zero centered to emphasize the trends exhibited. While TFT is less volatile, *it is also less accurate* as it fails to incorporate newly available information. By contrast, MQTransformer is both *less volatile* and *more accurate* when compared with MQ-CNN.

