# OpenReview forum: "MQTransformer: Multi-Horizon Forecasts with Context Dependent and Feedback-Aware Attention"
_ICLR.cc/2022/Conference — ICLR 2022 Submitted_

### Official Review · Reviewer_tUKV · 2021-10-29

**Correctness:** 3
**Technical Novelty And Significance:** 3
**Empirical Novelty And Significance:** 3
**Recommendation:** 6
**Confidence:** 4

**Main Review:**

Strengths

- A clear formal introduction to the problem statement and concise descriptions of the three contributions. The overall stricture and readability of the paper are good.

- Authors have attested their framework against state-of-the-art deep learning time series forecasting methods and evaluated using 5 benchmark datasets, including a one retail dataset, and four publicly available datasets. The benchmark methods and datasets used are well-established yardsticks in the multivariate forecasting (or global models where sets of many time series are available) domain. So, the experiments are strong and well empirically supported.

- The use of an ablation study to measure the effect of the proposed attention units and self-attention scheme.

- Have also reported the computational cost of the framework and compared against the current state-of-the-art (Lim et al., 2019)


Weaknesses

- Only the ablated based variants of MQTransformer are evaluated and compared on the large-scale retail dataset. What is the reason for excluding other forecasting benchmarks, such as DeepAR, TFT, MQ-RNN in Table 3 ?

- The formula used to calculate the quantile loss (quantile forecast error) is not defined. Please add this formula to the appendix or main text to improve the clarity of the error measures summarised in the tables.

- I also strongly recommend the authors to include state-of-the-art Informer framework [1] as a benchmark in this study. This is because, Informer framework also follows the self- attention architecture, thus can be directly comparable against the proposed MQTransformer.

 [1] Zhou, H., Zhang, S., Peng, J., Zhang, S., Li, J., Xiong, H. and Zhang, W., 2021, May. Informer: Beyond efficient transformer for long sequence time-series forecasting. In Proceedings of AAAI.


**Summary Of The Paper:**

This study proposes a decoder-encoder attention mechanism, a positional encoding mechanism and a decoder self-attention scheme to improve the forecast accuracy and minimise the excess variations in time series forecasting applications. As highlighted in the introduction section of the paper, both improving the forecast accuracy and minimising the unnecessary forecast volatility are important endeavours in the time series forecasting domain. Hence the motivations of this study are clear and strong. The proposed MQTransformer framework aims to obviate the limitations of MQ-Forecaster, which is also a state-of-the-art framework in the recent time series forecasting literature.

**Summary Of The Review:**

Given the significance of the three contributions, supported by the strong experimental setup (well established benchmarks and datasets) and empirical evidence (statistical significance), I am recommending to accept this paper.

---

> ### Author Response · Authors · 2021-11-23
> **Reply to Review**
>
>
> Thank you for taking the time and effort to review our paper -- we really appreciate it! We respond to several of the points below.
>
> > Only the ablated based variants of MQTransformer are evaluated and compared on the large-scale retail dataset. What is the reason for excluding other forecasting benchmarks, such as DeepAR, TFT, MQ-RNN in Table 3 ?
>
> Thank you for pointing out that this was not clearly explained in the manuscript. We compared the ablated variants only against MQ-CNN for several reasons:
>
> 1. For each benchmark, we would have needed to reimplement that model in order to scale it to the proprietary dataset. Since that is quite costly, we did not compare against MQ-RNN or DeepAR as MQ-CNN outperformed them on all public datasets. We also note that prior work [1] already compared against MQ-RNN and DeepAR (we reference this in the footnote on page 7) on the same proprietary dataset; both were out-performed by MQ-CNN.
> 2. TFT does not use forking sequences, and thus is difficult to train at scale (see the discussion of computational efficiency on page 9). It would have required extensive engineering work to make training TFT feasible on the proprietary dataset. Given how similar the Favorita task is, we decided it made more sense to perform the comparison to TFT on the public datasets only.
>
> > The formula used to calculate the quantile loss (quantile forecast error) is not defined. Please add this formula to the appendix or main text to improve the clarity of the error measures summarised in the tables.
>
> Thanks for bringing this to our attention. We have added the formula in Section 4.
>
> > I also strongly recommend the authors to include state-of-the-art Informer framework [1] as a benchmark in this study. This is because, Informer framework also follows the self- attention architecture, thus can be directly comparable against the proposed MQTransformer.
>
> Thank you for the suggestion -- we will consider doing this in a future revision. The Informer architecture did not exist at the time we did our work, which is why it was not included in the comparisons.
>
> ### References
> [1] Wen, R., Torkkola, K., Narayanaswamy, B. and Madeka, D. (2017). A multi-horizon quantile recurrent forecaster. In NIPS Time Series Workshop.

---

### Official Review · Reviewer_DW35 · 2021-11-02

**Correctness:** 4
**Technical Novelty And Significance:** 2
**Empirical Novelty And Significance:** 2
**Recommendation:** 5
**Confidence:** 4

**Main Review:**

This work applies Transformer architecture on forecasting tasks, and shows improvement over existing work, including the previous work that used Transformer for decoder. The overall framework follows the conventional Transformer design with minor additions, and therefore provides limited technical novelty. There is some room for improvement on the clarity, as some notations are not clearly defined or used to indicate different meanings at different paragraphs.


**Summary Of The Paper:**

This paper looks into applying Transformer with encoder-decoder architecture for forecasting. The work follows the previous work of multi-horizon quantile forecasts that predict quantile for each horizon. The framework applies a 1D-conv and a MLP network on time series feature to model positional information, and apply an attention head for each horizon. The authors also show that limiting the context of the attention provides better bias to improve the prediction quality. The authors also discussed that having access to past context allows reducing the volatility. The proposed framework showed some improvement on demand forecasting tasks.

**Summary Of The Review:**

The proposed approach shows some improvement compared to existing work, but it’s technical technical novelty is a bit limited.

---

> ### Author Response · Authors · 2021-11-23
> **Reply to Review**
>
> Thank you for taking the time to review our paper. If you have any more specific feedback, please let us know -- it would be greatly appreciated!

---

### Official Review · Reviewer_3wZQ · 2021-11-03

**Correctness:** 4
**Technical Novelty And Significance:** 3
**Empirical Novelty And Significance:** 2
**Recommendation:** 5
**Confidence:** 3

**Main Review:**

Pros:

a) Significant gains in forecast accuracy on the proprietary data, and on the public retail forecasting task

b) Including forecast volatility as part of the model leads to a significant reduction in excess volatility over the MQ-CNN model

Cons:

a) A large part of the empirical analysis is conducted on a proprietary dataset, and thus those results are not reproducible or verifiable.  Furthermore, in these results the authors mask absolute values and only present relative gains, making it even harder to assess performance.

b) For results on the public task, a more direct comparison with MQ-CNN model is needed as the proposed model aims to improve over this model.  Since previously published results from the MQ-CNN model are on the GEFCom2014 forecasting task, could this be used to compare the proposed model?

c) Ablation studies are needed to understand key contributors to the gains observed.  For instance, on the ‘retail’ public task, a 38% accuracy gain is observed over the TFT model, yet excess variability studies show that TFT model has a lower excess variability as compared to the MQ transformer model (Fig. 6 in appendix D.5).  This raises the question if less focus on reducing excess variability will lead to further accuracy gains on this task.

d) Reproducibility of results is questionable.  Implementation details are shared to some extent, but lack of code, large focus on internal datasets that can not be released and numbers on those can not be disclosed, and a lack of detailed ablations will make it difficult to reproduce in my opinion.

Minor edits:
* subscript ‘I’ missing from x^{(s)} after Eq. (1)
* what is s_i in the reformulated version of (1) on page 4
* ‘where c_{:t} denotes … but there is no c_{:t} on page 5


**Summary Of The Paper:**

The paper proposes an improvement over previously published MQ-forecaster model for multi-horizon forecasts in time series data.  The proposed approach aims to improve the forecast accuracy as well as achieve a lower variability in forecasts made at various times for a specific time point in the future (excess forecast variability).  A number of model improvements are proposed, including a new way of handling positional encoding (via learnt embedding of event indicators), separate encodings for different horizons, and decoder attention on previous forecast errors to allow decoder to be aware of excess forecast variability and reduce it.  Empirical evaluation is carried out for the task of demand forecasting on a large but proprietary dataset, and also for tasks of retail sales, electricity load, securities volatility, and traffic forecasting on publicly available datasets.

**Summary Of The Review:**

My rating is due to concerns about reproducibility (no code and bunch of results on proprietary dataset), and lack of comparison with previously published results for the MQ-CNN model which this paper aims to improve over.

---

> ### Author Response · Authors · 2021-11-23
> **Reply to Review**
>
> Thank you for taking the time to review our paper and provide feedback. We respond to several of the points below.
>
> ### Minor edits
> Thank you for bringing these to our attention, we've addressed them in the updated draft.
>
>
> ### Comparison with MQ-CNN
> > For results on the public task, a more direct comparison with MQ-CNN model is needed as the proposed model aims to improve over this model. Since previously published results from the MQ-CNN model are on the GEFCom2014 forecasting task, could this be used to compare the proposed model?
>
> We apologize for not making this comparison clearer in the manuscript. We do compare directly against MQ-CNN on four public datasets. These four datasets were chosen for comparison because prior work [1] comparing a wide range of models used these datasets.
>
> ### Ablation Studies
> > Ablation studies are needed to understand key contributors to the gains observed. For instance, on the ‘retail’ public task, a 38% accuracy gain is observed over the TFT model, yet excess variability studies show that TFT model has a lower excess variability as compared to the MQ transformer model (Fig. 6 in appendix D.5). This raises the question if less focus on reducing excess variability will lead to further accuracy gains on this task.
>
> Thank you for providing this feedback. Foster and Stine 2021 [2] showed the connection between *excess volatility* and forecast accuracy -- namely, any forecast that is overly volatile can be improved (made more accurate). Our decoder self-attention mechanism is inspired by the martingale filter in their work. TFT, while not overly volatile, is also less accurate than even the MQ-CNN baseline on the most complex public task (Favorita).
>
>
> ### Reproducibility
> Thank you for sharing this feedback. We appreciate the concerns regarding reproducibility and tried to provide sufficient detail to reproduce our results. Although we believe the details in the appendices should be enough to reproduce our results on the four public tasks, we do agree with your feedback that code would be better. Unfortunately, we did not receive permission to release code, data or absolute performance metrics (hence the relative ones we reported).
>
>
> ### References
> [1] Lim, B., Arik, S. O., Loeff, N. and Pfister, T. (2019). Temporal Fusion Transformers for Interpretable Multi-horizon Time Series Forecasting.
> [2] Foster, D. and Stine, R. (2021). Threshold Martingales and the Evolution of Forecasts.

---

> > ### Comment · Area_Chair_PBnQ · 2021-12-01
> > **Any further thoughts?**
> >
> > Thanks for the thoughtful review and response! Reviewer 3wZQ, any further thoughts after reading the response?

---

> > > ### Comment · Reviewer_3wZQ · 2021-12-01
> > > **Re: Any further thoughts?**
> > >
> > > I'd like to thank the authors for their response.  Having read the response and other discussions on this paper I'd like to keep my rating unchanged, I think the paper still remains below the threshold of acceptability.

---

### Decision · Program_Chairs · 2022-01-20

**Decision:**

Reject

**Comment:**

This paper proposes a number of improvements to the previously-published transformer-based MQ-forecaster model for multi-horizon forecasts on time series data. They show strong empirical improvements in terms of accuracy and excess forecast variability on a large proprietary dataset, as well as four public datasets. Concerns were raised about the relatively incremental changes to the MQ-forecaster model this work is based on, lack of ablations on public data and, relatedly, inability to reproduce results on the proprietary data.